# Differential Lipidomics, Metabolomics and Immunological Analysis of Alcoholic and Non-Alcoholic Steatohepatitis in Mice

**DOI:** 10.3390/ijms241210351

**Published:** 2023-06-19

**Authors:** Erika Dorochow, Nico Kraus, Nicolas Chenaux-Repond, Sandra Pierre, Anja Kolbinger, Gerd Geisslinger, Cristina Ortiz, Christoph Welsch, Jonel Trebicka, Robert Gurke, Lisa Hahnefeld, Sabine Klein, Klaus Scholich

**Affiliations:** 1Institute of Clinical Pharmacology, Goethe-University Frankfurt, 60590 Frankfurt, Germany; dorochow@med.uni-frankfurt.de (E.D.); chenaux-repond@med.uni-frankfurt.de (N.C.-R.); spierre@em.uni-frankfurt.de (S.P.); kolbinger@med.uni-frankfurt.de (A.K.); geisslinger@em.uni-frankfurt.de (G.G.); robert.gurke@itmp.fraunhofer.de (R.G.); hahnefeld@med.uni-frankfurt.de (L.H.); 2Center for Internal Medicine, Hospital of the Goethe University Frankfurt, 60323 Frankfurt, Germany; nico.kraus@kgu.de (N.K.); cristina.ortiz@kgu.de (C.O.); christoph.welsch@kgu.de (C.W.); 3Fraunhofer Institute for Translational Medicine and Pharmacology ITMP, 60596 Frankfurt, Germany; 4Fraunhofer Cluster of Excellence for Immune-Mediated Diseases CIMD, 60596 Frankfurt, Germany; 5Clinic for Internal Medicine B, Hospital of the University of Münster, 48149 Münster, Germany; jonel.trebicka@ukmuenster.de (J.T.); sabine.klein@ukmuenster.de (S.K.)

**Keywords:** non-alcoholic fatty liver disease, alcoholic fatty liver disease, lipid droplets, metabolomics, lipidomics

## Abstract

Non-alcoholic steatohepatitis (NASH) and alcoholic steatohepatitis (ASH) are the leading causes of liver disease worldwide. To identify disease-specific pathomechanisms, we analyzed the lipidome, metabolome and immune cell recruitment in livers in both diseases. Mice harboring ASH or NASH had comparable disease severities regarding mortality rate, neurological behavior, expression of fibrosis marker and albumin levels. Lipid droplet size was higher in NASH than ASH and qualitative differences in the lipidome were mainly based on incorporation of diet-specific fatty acids into triglycerides, phosphatidylcholines and lysophosphatidylcholines. Metabolomic analysis showed downregulated nucleoside levels in both models. Here, the corresponding uremic metabolites were only upregulated in NASH suggesting stronger cellular senescence, which was supported by lower antioxidant levels in NASH as compared to ASH. While altered urea cycle metabolites suggest increased nitric oxide synthesis in both models, in ASH, this depended on increased L-homoarginine levels indicating a cardiovascular response mechanism. Interestingly, only in NASH were the levels of tryptophan and its anti-inflammatory metabolite kynurenine upregulated. Fittingly, high-content immunohistochemistry showed a decreased macrophage recruitment and an increased polarization towards M2-like macrophages in NASH. In conclusion, with comparable disease severity in both models, higher lipid storage, oxidative stress and tryptophan/kynurenine levels were seen in NASH, leading to distinct immune responses.

## 1. Introduction

Alcoholic (AFLD) and non-alcoholic fatty liver disease (NAFLD) are among the major global diseases affecting 25–30% of the population in Western countries [1,2]. They span a spectrum of disease conditions ranging from fatty liver (steatosis) to alcoholic and non-alcoholic steatohepatitis (ASH and NASH, respectively) with increased probability for complications such as cirrhosis, acute-on-chronic liver failure (ACLF) and hepatocellular carcinoma. The histological features of ASH and NASH (i.e., hepatocyte ballooning, lobular inflammation) appear similar, suggesting partly common pathomechanisms. Here, especially the hepatic accumulation of triglycerides and phospholipids in lipid droplets is a hallmark for both diseases [2,3].

While in NASH the lipid accumulation is based on the excessive availability of lipids and carbohydrates in the food, in ASH, several mechanisms seem to add to the increased hepatic lipid storage. These include interruption of mitochondrial β-oxidation of fatty acids, increased hepatic de novo lipid synthesis as well as induction of lipolysis in adipocytes. As a consequence, circulating fatty acids increase, causing their subsequent hepatic accumulation [4,5]. Chronic alcohol uptake induces the expression of cytochrome p450 2E1 (CYP2E1) in the liver. CYP2E1 metabolizes alcohol to acetaldehyde and also generates high amounts of reactive oxygen species (ROS), which impair antioxidant mechanisms. Acetaldehyde is then converted to acetate, which can be utilized in several metabolic pathways, including the citric acid cycle, fatty acid oxidation, glycolysis and gluconeogenesis. It is not clear to which extent acetate directly contributes to hepatic fatty acid synthesis, because it can be rapidly secreted into the circulation, and it has been suggested that lipids released from peripheral adipocytes are responsible for increased hepatic lipid storage in ASH [3,6].

However, the inability of hepatocytes to metabolize the excess supply of lipids leads to the formation of lipid droplets in metabolic fatty liver diseases, to avoid lipotoxicity due to cellular overload of free fatty acids. At the beginning, lipotoxicity induces cell senescence in hepatocytes, which is marked by a cell cycle arrest while the cells remain metabolically active. Eventually, lipotoxicity-induced endoplasmic reticulum stress [7] and activation of the unfolded protein response [8] stimulate apoptotic pathways leading to apoptosis [9]. The subsequent release of damage-associated molecular patterns (DAMPs) induces the activation of inflammatory pathways in hepatocytes and subsequently hepatic stellate cells initiating the recruitment of immune cells and the generation of ROS. Additionally, in ASH, alcohol-induced CYP2E1 induction stimulates the transport of NADH into mitochondria, which is associated with increased electron leakage from the hepatocyte mitochondrial respiratory chain and additional ROS production [10]. In addition, the alcohol metabolite acetaldehyde has been reported to enhance lipotoxicity-induced ROS generation by causing mitochondrial functional impairment [11]. Ultimately, the combined stressors can cause hepatocyte apoptosis, which activates hepatic stellate cells and the immune systems and thereby promotes fibrosis in the affected areas. Due to the key role of lipotoxicity in fatty liver disease progression and the lipotoxicity-induced metabolic changes, we compared two mouse models for ASH and NASH with similar disease severity to identify disease-specific pathomechanisms in the lipidome, metabolome and the immunological response.

## 2. Results

### 2.1. Disease Severity in Mice Models for ASH and NASH Is Comparable

To identify distinct pathomechanisms in mice harboring ASH or NASH, we used previously established mouse models [12]. Here, CCl_4_ was administered twice weekly for seven weeks together with either ethanol in the drinking water to induce ASH or with a Western diet, a high-fat and cholesterol-enriched diet, to induce NASH. First, we compared both models to assess the strength of the disease in either model with several parameters to ensure comparable conditions in both models. In the physiological readouts, we found similar mortality rates between 15% and 10% in ASH and NASH, respectively (Figure 1A). The effect of the treatments on the neurological behavior differed significantly as compared to healthy mice but showed no differences between both models (Figure 1B). Additionally, albumin plasma levels decreased to a similar extent in both models (Figure 1C). Likewise, the platelet blood counts, which are predictive markers for disease severity, were decreased to comparable levels in both models (Figure 1D). However, AST/ALT ratios were slightly increased in ASH than in NASH (Figure 1E). Assessment of fibrosis based on histological Sirius Red staining as well as *collagen 1a1* (*Col1a1*) and *Acta2* (α2 smooth muscle actin) mRNA expression in hepatic stellate cells (HSCs) showed no differences between both disease models (Figure 1F–H). Loss of retinoid storage in HSCs is one of the characteristics of their activation, while the reason and the underlying mechanisms are not understood [13]. Interestingly, according to the autofluorescence of free retinoic acid in HSCs, the number of activated HSCs was around four times higher in NASH than in ASH (Figure 1I). Retinoic acid is generated from its non-fluorescent storage form as retinyl-ester. The synthesis ends with the depletion of retinyl-esters and the complete release of retinoic acid from HSCs. Accordingly, HSCs were either double-positive for retinoic acid autofluorescence and the HSC differentiation marker α-smooth muscle actin (αSMA) or single-positive for αSMA (Figure 1J). Because *Acta2* mRNA expression was similar in both models, the data suggest that disease progression was beginning to increase stronger in NASH than in ASH at the investigated time point. Because disease progression is connected with mitochondrial dysfunction, we determined the mRNA expression of SIRT1 and PGC-1α, which are downregulated in the event of mitochondrial dysregulation. No significant change in the mRNA expression of SIRT1 and PGC-1α were seen in ASH as compared to healthy control mice (Appendix A). However, in NASH mice, SIRT1 was significantly reduced as compared to healthy mice or mice with ASH, fitting with a mildly increased disease progression in NASH (Appendix A). Taken together, both models exhibit a similar disease severity after 7 weeks of treatment. The AST/ALT ratio suggests a slightly more advanced disease progression in ASH, whereas the retinoid content in HSCs indicates a stronger disease progression in NASH.

### 2.2. Major Changes in the Hepatic Lipid Content Are Influenced by the Specialized Diets 

Because hepatic lipid accumulation is a hallmark of fatty liver diseases, we determined the lipid content in livers from mice with ASH and NASH. Visualization of lipid droplets showed that the number of lipid droplets was higher in ASH than in NASH, while the size of the droplets was larger in NASH (Figure 2A–C). Lipid droplets store triacylglycerols more efficiently in large droplets through reduction of the lipid droplet surface area that is exposed to cytosolic lipases for lipolysis, suggesting a higher lipid content in NASH. Lipidomic analysis of 388 lipids showed clear differences in the principal component analysis (PCA) between livers from healthy mice and mice with ASH or NASH (Figure 2D). Detailed analysis of the lipids revealed that mainly triglycerides, phosphatidylcholines (PC) and lysophosphatidylcholines (LPC) were responsible for the separation between the models (Figure 2E). Nearly 80% of the lipids with the strongest changes as compared to livers from healthy mice were members of these three lipid families (Figure 2F; Appendix A), which all represent either lipids stored in lipid droplets or components of the lipid droplet membrane.

In NASH, triglycerides with fatty acids of lower chain lengths were increased up to 90-fold (Figure 3A). In contrast, the 20 lipids with the strongest increase in ASH were only upregulated 2–6-fold (Figure 3A). Analysis of the fatty acid composition of some of the strongest regulated triglycerides, PCs and LPCs revealed that all contained lauric acid (FA 12:0) and/or myristic acid (FA 14:0) in NASH (Figure 3B–D; Appendix A). This finding can be explained by the abundance of these two fatty acids in the Western diet, which NASH mice received (Figure 3E). Thus, the data illustrate that caution needs to be exercised during the interpretation of lipidomic comparisons between livers from mice with ASH and NASH, because the major differences in the lipid content in both models are based on their specific diets and cannot be easily attributed to underlying pathomechanisms. However, lipid signal mediators, such as prostanoids, are synthesized in response to physiological stimuli and therefore can give important insights in the pathomechanisms. Here, we found that hepatic levels of the inflammation mediator prostaglandin (PG) E_2_ and 6-keto-PGF_1α_, the stable metabolite of the vasoactive prostacyclin (PGI_2_), were significantly upregulated in ASH and NASH (Figure 3F,G). 

### 2.3. Metabolomics Reveal Disease-Specific Responses in ASH and NASH 

Because the excess availability of lipids induces alterations in the cellular lipid and energy metabolism, we compared ASH and NASH using a metabolomics approach. Metabolomic analysis of 130 metabolites in livers from untreated mice as well as mice with ASH or NASH showed clear differences between the three groups in a heatmap with unsupervised clustering and PCA (Figure 4A,B, Appendix A). A total of 43 metabolites were altered in ASH and 54 in NASH in comparison to livers from control mice, whereby 21 were similarly regulated in both diseases. Testing for known alterations in livers from mice with ASH and NASH showed that fucose was upregulated in ASH [14] and downregulated in NASH [15] (Figure 4C). In addition, bile acids were more strongly upregulated in NASH mice, which can be explained by the high-fat diet these mice received (Figure 4D,E).

Hepatic lipotoxicity is accompanied by cellular senescence, which is marked by a cell cycle arrest of hepatocytes, leading to a decreased nucleoside demand. Accordingly, in both fatty liver diseases, a similar downregulation of purine (Figure 5A,B) and pyrimidine (Figure 5A,C) nucleoside levels was observed. Notably, in NASH mice, metabolites of purine nucleosides were upregulated, suggesting their degradation due to increased cellular stress. Fittingly, the levels of the antioxidant ascorbic acid and its metabolite ascorbic acid 2-sulfate were significantly decreased in ASH and NASH (Figure 5D,E). However, levels of reduced glutathione were downregulated in NASH, whereas they were upregulated in ASH, suggesting an intact antioxidative system in the latter model (Figure 5F). 

Because the urea cycle is crucially involved in arginine metabolism, nitric oxide (NO) synthesis and ROS generation, we compared the hepatic arginine metabolism in mice with ASH and NASH. In the urea cycle, the levels of ornithine, N-acetyl-ornithine and citrulline increased in NASH as compared to healthy mice (Figure 6A,B). This pattern is typical for a NO-induced blockage of the citrulline-metabolizing enzyme arginosuccinate synthase [16]. In contrast, the levels of the non-proteinogenic amino acids homoarginine, ornithine and N-acetyl-ornithine were increased in ASH (Figure 6A,B). Homoarginine is an alternative substrate for NO synthases (NOS), which is connected to improved cardiovascular function and, therefore, is believed to be produced in the endothelium [17]. However, it should be noted that the disease-induced portal pressure was similar in both models, which puts a relevant vascular effect in question (Figure 6C).

### 2.4. Hepatic Immune Responses Differ between ASH and NASH 

Furthermore, the tryptophan metabolism was selectively upregulated in NASH leading to increased levels of tryptophan, kynurenine and Indoleacrylic acid (Figure 7A,B). Because kynurenine and its metabolites have immunosuppressive effects [18], we determined the immune response in these mice. Therefore, we used the MELC technology for high-content immunohistochemistry, which was used to visualize 30 antibodies (Appendix A) describing the major immune cell types [19]. To determine immune cells in the liver tissue, the field of visions were chosen to cover parts of the liver, which contained activated HSCs according to their autofluorescence. Because no activated HSCs were observed in livers from healthy mice, we randomly chose similar tissue areas around vessels. For quantitative assessment of cells, a segmentation mask was generated based on nuclear staining (propidium iodide) and was restricted to the liver tissue excluding the vascular space. Single-cell phenotyping was performed using PhenoGraph analysis, followed by identification and quantification of cell clusters representing the different immune cell types as described previously [19,20]. Both disease models had similar numbers of eosinophils, dendritic cells, T-cells and platelets, while B-cells and NK or NKT cells were not detected (Figure 7C). The number of neutrophils was significantly increased in NASH but not in ASH as compared to healthy mice (Figure 7C). Macrophage numbers within the tissue were significantly increased in both models, whereas their number was roughly twofold higher in ASH than in NASH (Figure 7C). This finding is consistent with previous reports, which show that kynurenine suppresses monocyte and macrophage activation in mice [21,22]. CD86^+^ proinflammatory M1-like macrophages were detected in neither model (Figure 7D; Appendix A). More importantly, the proportion of anti-inflammatory CD206^+^ macrophages increased in NASH (Figure 7E), which is again in accordance with the high hepatic kynurenine levels in NASH, because kynurenine has been shown to support polarization of M2-like macrophages [23]. 

## 3. Discussion

We used metabolomic, lipidomic and immunologic approaches to investigate common and distinct pathomechanisms of ASH and NASH. 

As animal models, co-treatment of mice with ethanol and CCl4 for ASH induction or a Western diet and CCl4 for NASH induction was chosen [12]. Mice receiving CCl4 alone showed, in absence of fatty liver disease, an induction of liver fibrosis as well as reduced body and liver weight. The combination of CCl4 with a Western diet enhances the full spectrum of fibrosis, steatosis and inflammation including collagen accumulation, hepatic stellate cell activation and hepatic steatosis [12]. We compared the transcription levels of markers of fibrosis, steatosis, inflammation and proliferation in CCl4/ethanol and CCL4/Western diet mouse models. Transcript correlation and PC analysis matched well between the CCl4/ethanol model and a human alcoholic liver cirrhosis cohort as well as the CCl4/WD model and a human NAFLD cohort, confirming etiology-specific modeling in the mice models [12]. More recent publications were able to correlate effects of janus kinase 2 inhibitors [24] as well as the role of macrophage-inducible C-type lectin (mincle) [25] in these models with results from human cohorts.

The two animal models for ASH and NASH had overall similar disease severity, although distinct lipidomic, metabolomic and immunologic features between the models were seen. For example, while the frequency of lipid droplets was higher in ASH, lipid droplets were larger in NASH. The lower frequency in NASH might be due to the larger size of the droplets in NASH, which reduces the distance between lipid droplets, thereby masking the real frequency of droplets in the microscopic images. Additionally, fusion of lipid droplets may take place to form larger lipid droplets. These larger droplets store triacylglycerols more efficiently through reduction of the lipid droplet surface area. However, the larger size of the droplets represents a higher lipid content in NASH than in ASH. This increased lipid storage is owed to the nature of the NASH model, which is based on a Western diet providing a massive excess of fatty acids. The lesser lipid overload in ASH together with the similar disease severity in ASH and NASH underlines that the pathomechanism in ASH is based only in part on lipid toxicity but also on multiple other toxic effects of alcohol intake [3,11]. Likewise, the higher hepatic lipid storage in NASH in combination with otherwise similar disease severity markers in both models underlines the stronger dependence on lipid toxicity as major cause of liver damage in NASH. 

HSCs are the main matrix-producing mesenchymal cells in the liver and, therefore, take a key role in fibrogenesis. Their activation is initiated by oxidative stress or inflammatory signals [13,24], which is in line with oxidative stress caused by the excess lipid amounts. HSC activation leads to the release of retinoic acid from its retinyl-ester storage form. This increased autofluorescence in HSCs is intermittent and ends with the depletion of the retinyl-ester and the complete release of the free retinoic acid from the cells. While the exact chain of events leading to the release of retinoic acids is not well understood, it is accepted as a hallmark for HSC activation [13,24]. Based on the autofluorescence of retinoid acid, the number of activated HSCs was higher in NASH than in ASH. Notably, despite the increased HSC activation in NASH, fibrosis was similar in ASH and NASH as shown by expression of the HSC differentiation markers *ACTA2* and *Col1a1* as well as the Sirius Red staining. Thus, the increased HSC activation in NASH signals a faster progression of fibrosis in NASH at this time, which did not yet manifest in differences in fibrosis. 

The increased lipid droplet size in hepatocytes in NASH as compared to ASH shows a stronger deposition of lipids in lipid droplets in NASH. This was mirrored in the lipidomic analysis by the very strong increases (up to 90-fold) in certain TG, PC and LPC species. These strong increases are at least in part due to the incorporation of fatty acids, which are either exclusively (myristate) or abundantly (laureate) available in the Western diet. Because these fatty acids were incorporated in lipids (i.e., PCs, LPCs and mainly triglycerides), which are typically found in lipid droplets, it seems that they are converted and stored in lipid droplets after their uptake in hepatocytes. The changes observed in the lipidomics analysis are more moderate in ASH and show a greater heterogeneity of upregulated lipid species. This heterogeneity is expected for lipids, which are newly synthesized from alcohol-derived acetate or are derived by an alcohol-induced release from peripheral adipocytes [4,5]. It should be noted that the strong diet-specific increase in many lipids in NASH makes it difficult to determine whether specific lipids are linked to the pathomechanisms or are only increased to a higher availability based on the Western diet. Therefore, the differentiation between merely diet-based changes and alterations connected with the pathomechanisms needs to be taken into consideration when interpreting comparative lipidomic analyses in animal models or human studies comprising cohorts with different eating habits. 

The metabolomics analysis of the livers from mice with ASH and NASH revealed several similarities and model-specific alterations, such as in the nucleoside and amino acid metabolism. In both models, a similar decrease in nucleoside levels was observed, which is in accordance with hepatic cellular senescence occurring during lipid- and alcohol-induced cellular stress [25]. However, the levels of nucleoside catabolites were only increased in NASH, suggesting a stronger cellular stress in this model. Fittingly, the antioxidants ascorbic acid and glutathione were significantly decreased in NASH, whereas in ASH only ascorbic acid levels were decreased. The stronger oxidative stress in NASH might be also the reason for the aforementioned higher activation frequency of HSCs. 

In accordance with increased oxidative stress, the alterations in the urea cycle suggest an increased production of NO in NASH. Here, ornithine and citrulline levels increased, which is typical for increased NO synthesis. NO inhibits the citrulline-metabolizing enzyme arginosuccinate synthase, leading to the accumulation of citrulline and ornithine [16]. In ASH, we found the levels of L-homoarginine and ornithine to be increased. L-Homoarginine is generated from lysine and arginine by the arginine:glycine amidinotransferase (AGAT) and serves as a substrate for NOS to produce NO and ornithine [26]. Although the liver has high L-homoarginine levels and unusually high AGAT activity, the physiological function of L-homoarginine in healthy and diseased livers is not clear [27]. However, in healthy blood vessels, L-homoarginine is a substrate for the endothelial nitric oxide synthase (eNOS) to produce NO and causes vasodilatation. Indeed, epidemiological studies show that low L-homoarginine plasma levels are inversely correlated with cardiovascular and all-cause mortality [28,29]. Thus, L-homoarginine might be synthesized in ASH mice as a response to the increasing portal hypertension. However, the reasons for a specific upregulation of this pathway in ASH are unclear, because hepatic levels of 6-keto PGF_1α_, the stable metabolite of the strong vasodilator prostacyclin, were equally upregulated in ASH and NASH.

Another amino acid metabolic pathway specifically altered in NASH involves tryptophan metabolism. Here, we observed an upregulation of tryptophan and its metabolite kynurenine. Hepatocytes are the only cells with high tryptophan 2,3-dioxygenase (TDO) activity [30], which is one of the two enzymes able to convert tryptophan to kynurenine. TDO is an enzyme with low affinity for tryptophan, allowing it to remain active even at the high tryptophan levels we observed in NASH [31]. Kynurenine and its metabolites suppress the activity of natural killer cells [32] as well as dendritic cells, monocytes, and macrophages in mice [21,22]. Fittingly, we observed that the recruitment of macrophages to the hepatic tissue was around 50% lower in NASH than in ASH. Additionally, the proportion of anti-inflammatory M2-like macrophages doubled in NASH, further supporting an anti-inflammatory role of kynurenine in NASH, because it promotes M2-like macrophage polarization [23]. At the same time, a significant recruitment of neutrophils was only seen in NASH, possibly compensating the decreased macrophage response. Notably, leukocytes can transport platelets in inflamed tissues where the platelets promote inflammatory processes [33,34,35]. We found similar platelet numbers in liver tissue from ASH and NASH mice, indicating that neutrophils replace macrophages in the hepatic platelet transport. Once in the tissue, platelets can be activated and suppress the expression of markers for M2-like macrophages [34] and activate HSCs through a variety of mediators [33].

Taken together, we found that ASH and NASH share several alterations of the lipid and metabolite responses, but that they also have distinct pathomechanistic features, which become evident in metabolomic and immunologic approaches. Here, especially the dysregulation of the urea cycle and the tryptophan metabolism shows clear differences between the two models for fatty liver disease. These differences are reflected by physiological responses including the portal pressure and the immunological answer to the cellular stress, although it should be noted that future studies are needed to add experimental data to support the here-reported data.

## 4. Materials and Methods

### 4.1. Mice

Male C57BL/6J mice (12 weeks) were purchased from Charles River (Laboratories Research Model and Services Germany, Sulzfeld, Germany). All animals received human care according to the criteria outlined in the EU regulations on animal research (2010/63/EU). The animals had free access to food and water and were housed in climate- (22 ± 0.5 °C; 50% humidity) and light-controlled rooms (light from 6.00 a.m. to 6.00 p.m.). All experiments were performed in accordance with the German animal protection and welfare law and the guidelines of the animal care facility at the Hospital of the Goethe University Frankfurt. 

### 4.2. Induction of Chronic Liver Disease

To induce ASH or NASH, previously established mouse models [11] were used, whereby CCl_4_ was administered by intraperitoneal injections [36]. Liver fibrosis was induced by CCl4 injection (i.p) 2 µL/g body weight (CCl4/Corn oil = 1:3, *v*:*v*) two times per week for seven weeks. CCl_4_ injections were combined with ethanol in the drinking water (4% during week 1, 8% during week 2 and 16% until animals were sacrificed) and normal chow (SAFE 150 from SAFE Complete Care Competence, Rosenberg, Germany) to induce ASH. NASH was induced by CCl_4_ injections in combination with a high-fat cholesterol-rich Western diet (WD) (diet #S0279-S011, Ssniff Spezialdiäten GmbH, Soest, Germany). Control mice received no CCl_4_ administrations and were fed normal chow.

### 4.3. Brain Injury Assessment

To register mental disorders due to liver cirrhosis, the Revised Neurobehavioral Severity Score for rodents was performed. This contains a continuous series of specific, sensitive and standardized observational tests that evaluate balance, motor coordination and sensorimotor reflexes in mice, as described previously [37].

### 4.4. Blood Parameters

To evaluate liver function, Aspartate Aminotransferase (AST), Alanine Aminotransferase (ALT) and Albumin (ALB) were analyzed in Li-Heparin plasma (S-Monovette LH, Sarstedt, Nürnbrecht, Germany) using the Fuji Drychem NX 500i (Scilvet, Weinheim, Germany). Li-Heparin blood was analyzed using the scil Vet abc Plus+ (Scilvet, Weinheim, Germany) to examine the platelet count.

### 4.5. Sirius Red Staining

For histology, liver samples of the left lobe were fixed in 4% formalin for 24 h, embedded in paraffin and sectioned slides (3–5 µm) were stained with 0.1% saturated picric acid (Chroma, Münster, Germany) to evaluate liver fibrosis. Stainings were captured using a Keyence BZ-X810 microscope (Keyence, Osaka, Japan) and quantified via open-source ImageJ software (V.1.51j8; National Institutes of Health, Bethesda, MD, USA). 

### 4.6. Quantitative Real-Time PCR

RNA isolation and quantitative polymerase chain reaction (qPCR) were performed as described previously [12]. Briefly, total RNA was isolated using a common Trizol (TRIzol Reagent, ambion, Carlsbad, CA, USA)-based protocol. cDNA synthesis was performed by the ImProm-II Reverse Transcription System (Promega, Madison, WI, USA). TaqMan gene expression assays (Thermo Fisher Scientific, Waltham, MA, USA) were used for qPCR according to the manufacturer’s protocol on a StepOnePlus Real-Time PCR System (Applied Biosystems, Foster City, CA, USA) with the oligonucleotides *Acta2* (Mm00725412_s1) and *Col1a1* (Mm00801666_g1) (Fisher Scientific GmbH, Schwerte, Germany). Experiments were carried out in duplicates. Gene expression was calculated by the 2^−ΔΔCt^ method and results were normalized to 18S rRNA expression as an endogenous control.

### 4.7. Portal Pressure Measurement

Mice were anaesthetized with a mixture of ketamine (Ketaset, 100 mg/mL; Zoetis, Berlin, Germany) and xylazine (Xylazin, 20 mg/mL; WDT, Garbsen, Germany). To evaluate portal hypertension, the portal pressure was measured in each mouse. The portal vein was cannulated with a 25-gauge safety-multifly needle (Sarstedt, Nümbrecht, Germany), fixated with a vascular clamp and connected to a highly sensitive pressure transducer. The external zero reference point was placed at the midportion of the animal. The measurements were recorded for at least 1 min on a multichannel computer-based recorder using PowerLab 8/35 and the LabChart Software (Powerlab; ADInstruments, Dunedin, New Zealand). The final value for portal pressure of one biological replicate was determined as the mean in the recorded phase.

### 4.8. High-Content Multiplex Imaging

Multi-epitope-ligand cartography (MELC) is an automated immunohistological imaging method that can be used to visualize unlimited numbers of antibodies on the same sample [33,34,35]. Briefly, liver tissue sections were taken at 10 µm thickness on silanized cover slips, fixed in 4% paraformaldehyde in PBS for 15 min, permeabilized with 0.1% Triton X100 in PBS for 15 min and blocked with 3% BSA in PBS for 1 h. The tissue was placed on the stage of a Leica DMI8 (Leica Microsystems, Wetzlar, Germany) and a picture was taken. Then, in an automated procedure, the sample was incubated with bleachable fluorescence-labeled antibodies and washed with PBS. Afterwards phase-contrast and fluorescence signals were imaged by a Leica DFC9000 GT camera (Leica Microsystems, Wetzlar, Germany). A bleaching step was performed to delete fluorescence signals and the post-bleaching image was recorded. Then, the next antibody was applied and the process was repeated. The antibodies used are listed in Appendix A.

### 4.9. Image Analysis

Image analysis was performed as previously described [35]. Briefly, the post-bleaching images were subtracted from their following fluorescence image. First, all greyscale antibody channel images were processed using ImageJ v1.52 (NIH, Bethesda, MD, USA) to diminish noise and background fluorescence. Then, Cell Profiler (v3.1.9) [38] was used for generation of a cell mask for single-cell segmentation using the images for propidium iodide (cell nuclei) and artifacts were removed for further analyses if necessary. The segmentation mask was imported into histoCAT (v1.76) (20) with the corresponding antibody channel images. All images, excluding the images used for single-cell mask generation, were z-score normalized and used for PhenoGraph analysis [39] as implemented in histoCAT. The PhenoGraph defines cell clusters based on single-cell mask and marker colocalization (k set to 30). The PhenoGraph clusters were classified as distinct cell types based on their marker expression, and this classification was used to quantify cells in the images. To determine the relative number of cells per cell type, the number of objects per cluster was normalized against the total number of objects in the cell mask.

### 4.10. Lipid Droplet Quantification

The tissue was treated as in the multi-epitope-ligand cartography experiments. The tissue was then incubated with LipidSpot (1:1000 in 1% BSA/PBS) for 30 min and then stained with DAPI for 5 min. For each tissue section, three fields of view were examined with a Zeiss Axio Observer.Z1 (Zeiss, Oberkochen, Germany) and images were taken in the FITC and DAPI channel using the Zeiss AxioCam. Afterward, Cell Profiler (version 3.1.9) was used to perform illumination correction and to determine the size, area and number of lipid droplets with ImageJ v1.52 (National Institutes of Health, Bethesda, MD, USA).

### 4.11. Profiling of Lipids and Polar Metabolites 

For homogenization, tissue pieces were covered with varying volumes of pre-cooled extraction solution (25% ethanol + 10 µM indometacin), depending on the individual tissue weights. Samples were homogenized by wet grinding using a Precellys 24-Dual tissue homogenizer coupled with a Cryolys cooling module (both Bertin Technologies, Montigny-le-Bretonneux, France).

For extraction, liver homogenates were diluted using pre-cooled extraction solution to tissue concentrations of 0.05 mg/µL. Lipids and polar metabolites were extracted as previously described via liquid–liquid extraction using 20 µL of tissue homogenate, corresponding to 1 mg of tissue material [40,41]. Samples were kept in ice water during processing and the extracted and dried samples were stored at <−70 °C until analysis. Just before analysis, the dried organic layer was reconstituted with 100 µL of methanol while the dried water layer was resuspended in 100 µL of 50% acetonitrile for the profiling of lipids and polar metabolites, respectively. QC samples were created from equivalent volumes of homogenized samples that were processed the same day.

For the profiling of lipids and polar metabolites, analysis was conducted on a Vanquish Horizon UHPLC system coupled with an Orbitrap Exploris 480 mass spectrometer (both Thermo Fisher Scientific, Dreieich, Germany) as previously described [41]. Briefly, chromatographic separation for lipids was performed using a Zorbax RRHD Eclipse Plus C8 column (50 mm × 2.1 mm ID, 1.8 µm particle size) with a pre-column of the same type (both Agilent Technologies, Waldbronn, Germany). A 14 min binary gradient was applied using water + 0.1% FA + 10 mM ammonium formate (eluent A) and ACN:IPA (2:3, *v*/*v*) + 0.1% FA (eluent B) (Appendix A).

Polar metabolites were chromatographically separated on a SeQuant ZIC-HILIC column (100 mm × 2.1 mm internal diameter, 3.5 µm particle size) with a corresponding pre-column (both Merck, Darmstadt, Germany). A binary gradient elution was applied using water + 0.1% FA (eluent A) and ACN + 0.1% FA (eluent B) (Appendix A). 

The UHPLC-HRMS system was operated via XCalibur software v4.4 and Tune Application 3.1 (both Thermo Fisher Scientific, San Jose, CA, USA). Lipids and polar metabolites were identified using Compound Discoverer 3.1 (Thermo Fisher Scientific, San Jose, CA, USA) with the mzCloud library and the LipidBlast VS68 positive and negative libraries. The semi-quantitative data were then evaluated in TraceFinder software v5.1 (Thermo Fisher Scientific, San Jose, CA, USA). A heated electrospray ionization (H-ESI) source was used for the ionization of polar metabolites. MS data were acquired in full scan mode using a resolution of 120,000. MS2 spectra were collected in a data-dependent manner (ddMS2) with a resolution of 15,000 and a total cycle time of 600 ms.

### 4.12. Oxylipine Analysis

Prostanoid levels were measured as previously described [41] with slight adaptations for the analysis of tissue samples. Briefly, tissues were homogenized by wet bead milling using zirconium oxide beads in 2 mL reinforced tubes (Bertin Instruments, Frankfurt, Germany) and a Precellys 24-Dual with a Cryolys cooling module (Bertin Instruments) as previously described [42]. After weighing the tissue samples, we added weight-dependent volumes of pre-cooled 25/75% (*v*/*v*) ethanol and water supplemented with 10 µM indomethacin to a final tissue concentration of 0.1 mg/µL. We then extracted 200 µL of this homogenate using a combination of protein precipitation and solid-phase extraction for analysis by liquid chromatography-tandem mass spectrometry (LC-MS/MS) (Appendix A).

### 4.13. Data Analysis and Statistics

Statistical analyses were performed using Prism V.9.5 (GraphPad, San Diego, CA, USA). Determination of statistically significant differences in all experiments was conducted with one- or two-way analysis of variance (ANOVA) followed by post hoc correction. Comparison of two groups was performed by Student’s *t*-test with Welch’s correction. Where indicated, a nonparametric Mann–Whitney U test was performed. All n-numbers describe biological replicates. PCA scores und loadings plots as well as clustered heatmaps were created using the MetaboAnalyst 5.0 web-based platform.

## Figures and Tables

**Figure 1 ijms-24-10351-f001:**
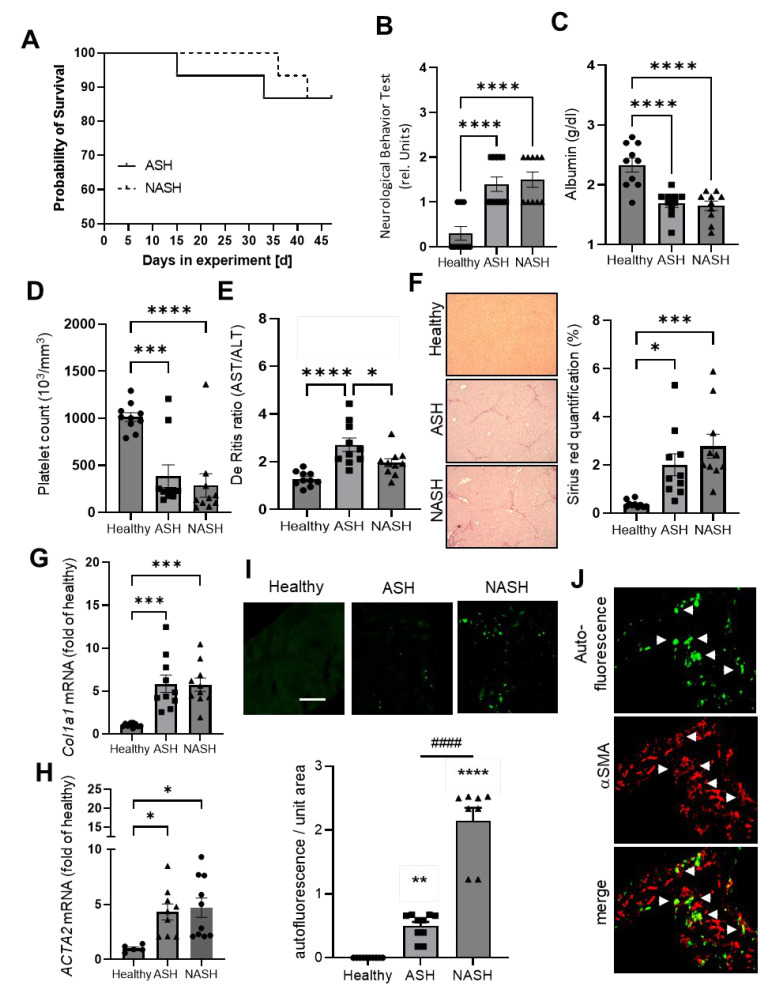
Mice models for ASH and NASH have similar disease severity. (**A**) Survival rate of mice during 7 weeks of treatment to induce ASH and NASH. Data are *n* = 15. (**B**–**E**) Neurological behavior (panel **B**), albumin level (panel **C**), platelet count (panel **D**), and de Ritis ratio (panel **E**) in healthy mice and 7 weeks after induction of ASH or NASH. Data are shown as mean ± SEM (*n* = 10). One-way ANOVA, Tukey post hoc test, * *p* < 0.05, *** *p* < 0.001, **** *p* < 0.0001. (**F**) Representative images and quantification of Sirius Red staining of liver tissue from healthy mice and 7 weeks after induction of ASH or NASH. Data are shown as mean ± SEM (*n* = 10). One-way ANOVA, Tukey post hoc test, * *p* < 0.05, *** *p* < 0.001. (**G**,**H**) Col1A1 (panel **G**) and Acta2 (panel **H**) mRNA in liver tissue from healthy mice and 7 weeks after induction of ASH or NASH. Data are shown as mean ± SEM (*n* = 10). One-way ANOVA, Tukey post hoc test, * *p* < 0.05, *** *p* < 0.001. (**I**) Representative images and quantification of autofluorescence in HSCs of healthy mice and 7 weeks after induction of ASH or NASH. Data are shown as mean ± SEM (*n* = 6). One-way ANOVA, Tukey post hoc test, ** *p* < 0.01, **** *p* < 0.0001 vs. healthy, #### *p* < 0.0001 ASH vs NASH. The size bar represents 300 µm. (**J**) Representative images of αSMA expression and retinoic acid autofluorescence in HSCs 7 weeks after induction of NASH. White arrowheads depict examples of colocalization of autoflourescent cells and αSMA expression.

**Figure 2 ijms-24-10351-f002:**
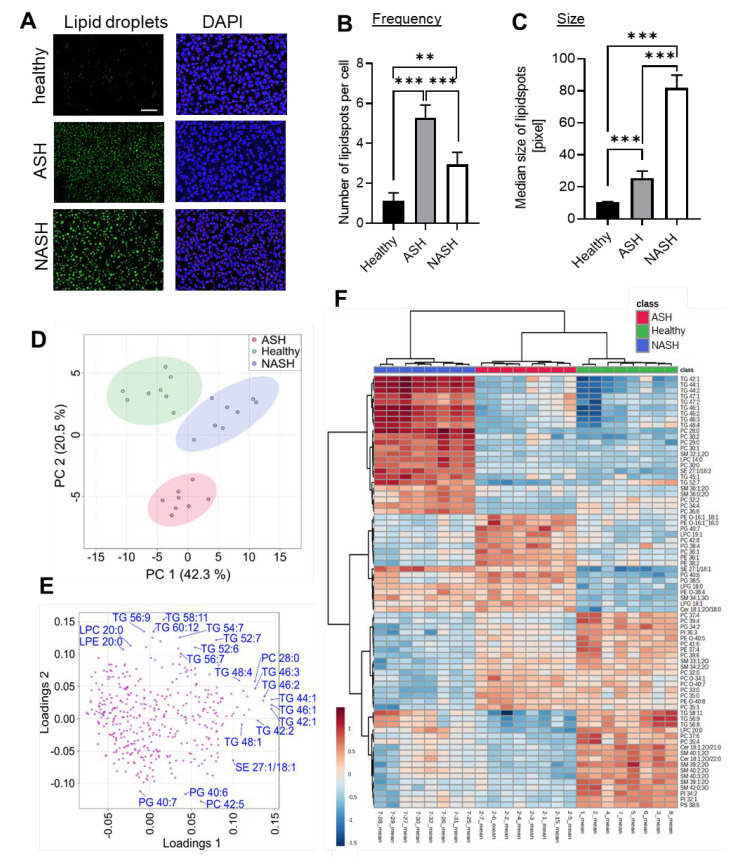
Lipid droplets in livers from mice with ASH and NASH. (**A**–**C**) Representative images of staining for lipid droplets and nuclei (DAPI) (panel **A**) and quantification of frequency (panel **B**) and size (panel **C**) of lipid droplets in liver tissue from healthy mice and 7 weeks after induction of ASH or NASH. The white bar represents 100 µm. Data are shown as mean ± SEM (*n* = 6). One-way ANOVA, Tukey post hoc test, ** *p* < 0.01, *** *p* < 0.001. (**D**) Principal component analysis (PCA) scores plot of lipidomics data for liver samples obtained from healthy mice and mice with induced ASH or NASH after 7 weeks. (**E**) Loadings plot illustrating the contribution of analyzed lipids to the corresponding PCA scores plot. A selection of lipids that are located at the outermost parts of the plot are labeled, as they possess the greatest contributions to the group separations in the PCA scores plot. (**F**) Heatmap of the top 75 lipids with the strongest changes based on ANOVA for the comparison of liver tissue obtained from healthy mice and mice with induced ASH or NASH after 7 weeks. Samples and lipids were clustered based on Ward’s method with Euclidean distance measure.

**Figure 3 ijms-24-10351-f003:**
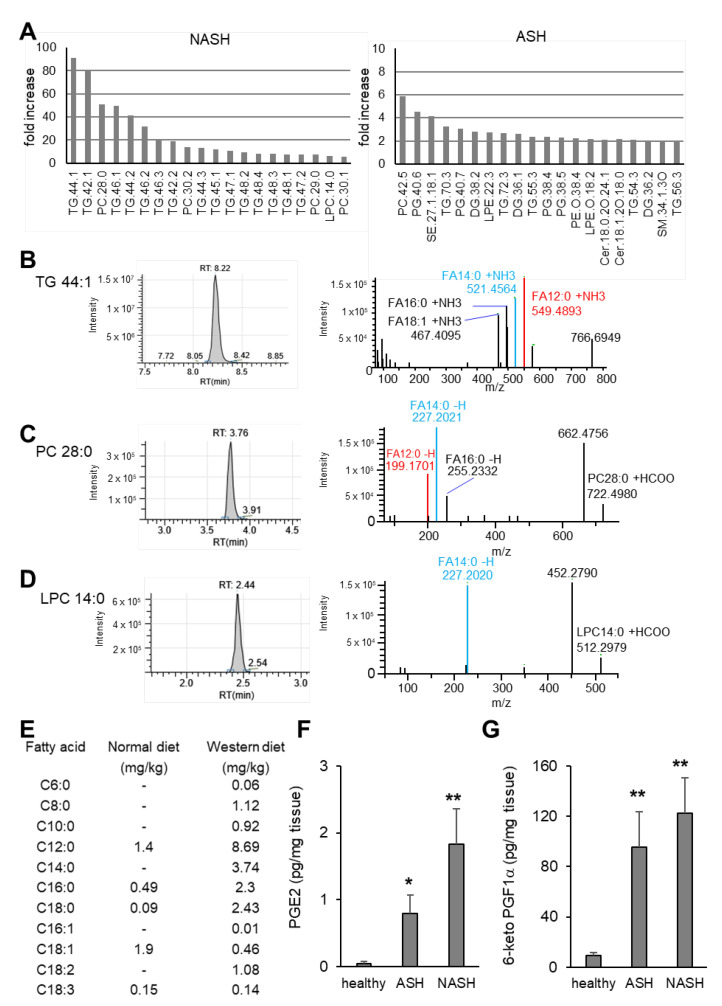
Diet-specific changes in lipidomics. (**A**) Fold increase of the 20 lipids with the strongest upregulation in NASH (left) and ASH (right) as compared to healthy controls. (**B**–**D**) Representative chromatograms and MS^2^ spectra of NASH samples for TG 44:1 (panel **B**), PC 28:0 (panel **C**) and LPC 14:0 (panel **D**). *m*/*z* = 549.4893 and *m*/*z* = 521.4564 correspond to the fragments that underwent the neutral losses of FA 12:0 + NH3 and FA 14:0 + NH3, respectively (panel B). *m*/*z* = 199.1701 and *m*/*z* = 227.202 correspond to the hydrogen-loss fragments of FA 12:0 and FA 14:0, respectively (panels **C**,**D**). *m*/*z* signals corresponding to FA 12:0 and FA 14:0 are indicated in red and blue, respectively. (**E**) Fatty acid composition of normal and Western diet. (**F**,**G**) PGE_2_ (panel **F**) and 6-keto PGF1α (panel **G**) in liver tissue from healthy mice and 7 weeks after induction of ASH or NASH. Data are shown as mean ± SEM (*n* = 8). One-way ANOVA, Tukey post hoc test, * *p* < 0.05, ** *p* < 0.01. FA: fatty acid; LPC: lysophosphatidylcholine; NH3: ammonia; PC: phosphatidylcholine; TG: triglyceride.

**Figure 4 ijms-24-10351-f004:**
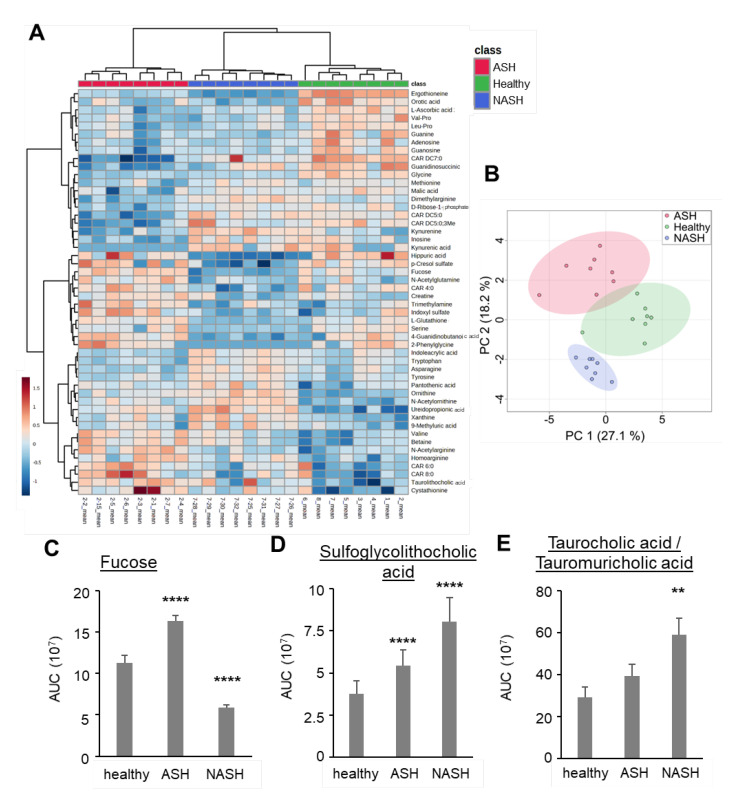
Metabolomics analysis of liver tissue from healthy mice and mice with induced ASH and NASH after 7 weeks. (**A**) Heatmap of the top 50 metabolites with the strongest changes based on ANOVA for the comparison of liver tissue obtained from healthy mice and mice with induced ASH or NASH after 7 weeks. Samples and metabolites were clustered based on Ward’s method with Euclidean distance measure. (**B**) Principal component analysis (PCA) scores plot of metabolomics data for liver tissue obtained from healthy mice and mice with ASH or NASH after 7 weeks. (**C**–**E**) Levels of fucose (panel **C**), sulfoglycolithocholic acid (panel **D**) and taurocholic acid/tauromuricholic acid (panel **E**) in liver tissue from healthy mice and mice with induced ASH or NASH after 7 weeks. Data are shown as mean ± SEM (*n* = 8). One-way ANOVA, Tukey post hoc test, ** *p* < 0.01, **** *p* < 0.0001.

**Figure 5 ijms-24-10351-f005:**
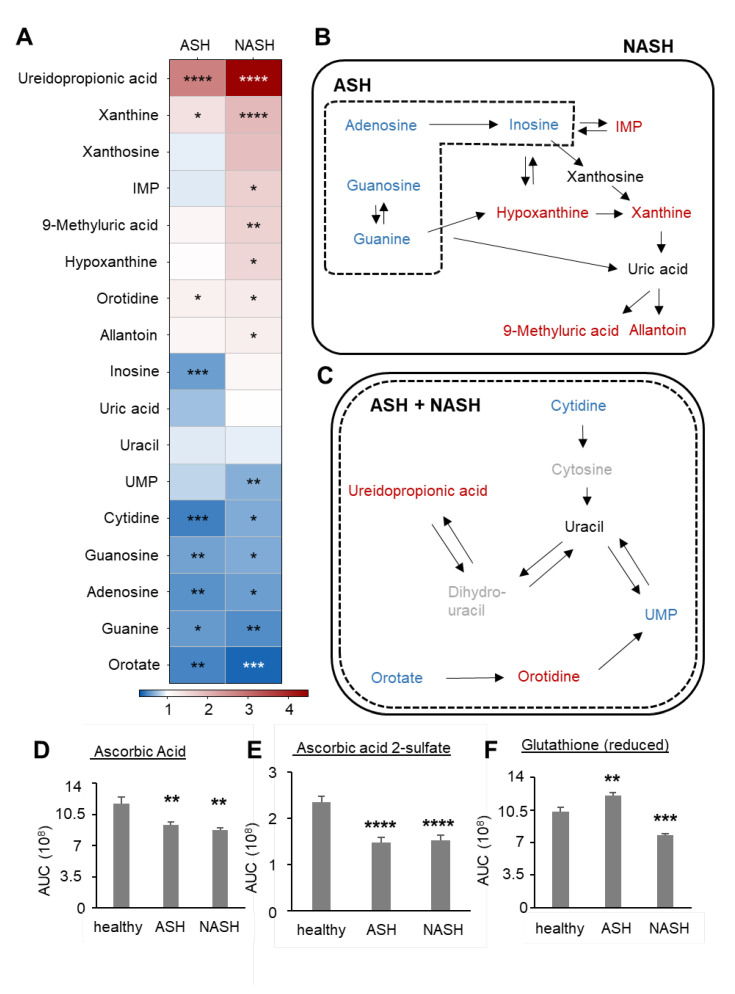
Disease-specific alterations in the nucleoside metabolism. (**A**) Heatmap of the detectable nucleosides and their metabolites in liver tissues 7 weeks after induction of ASH or NASH. Data are shown as fold changes compared to healthy mice (*n* = 8). Student’s *t*-test vs. healthy mice, * *p* < 0.05, ** *p* < 0.01, *** *p* < 0.001, **** *p* < 0.0001. (**B**,**C**) Schematic illustration of the purine (panel **B**) and pyrimidine (panel **C**) nucleoside metabolic pathways. Metabolites in the dashed and continuous boxes are regulated in ASH and NASH, respectively. Metabolites shown in blue are significantly downregulated, whereas metabolites in red are significantly upregulated. Metabolites indicated in black show no significant changes and metabolites in grey were not detected. (**D**–**F**) Levels of ascorbic acid (panel **D**), ascorbic acid 2-sulfate (panel **E**) and reduced glutathione (panel **F**) in liver tissue from healthy mice and mice with induced ASH or NASH after 7 weeks. Data are shown as mean ± SEM (*n* = 8). One-way ANOVA, Tukey post hoc test, ** *p* < 0.01, *** *p* < 0.001, **** *p* < 0.0001. IMP: inosine monophosphate; UMP: uridine monophosphate.

**Figure 6 ijms-24-10351-f006:**
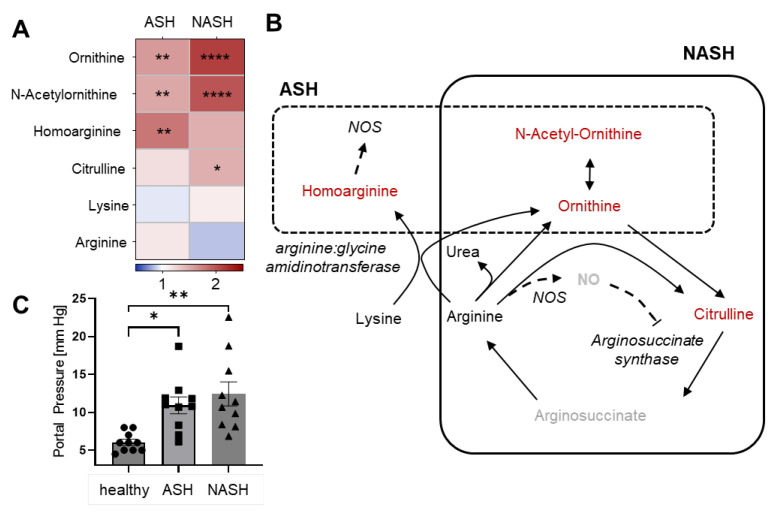
Disease-specific alterations in the urea cycle metabolism. (**A**) Heatmap of the detectable metabolites of the urea cycle in liver tissues 7 weeks after induction of ASH or NASH. Data are shown as fold changes compared to healthy mice (*n* = 8). Student’s *t*-test vs. healthy mice, * *p* < 0.05, ** *p* < 0.01, **** *p* < 0.0001. (**B**) Schematic illustration of the urea cycle. Important enzymes are shown in italic. Metabolites in the dashed and continuous boxes are regulated in ASH and NASH, respectively. Metabolites shown in red are significantly upregulated, whereas metabolites indicated in black show no significant changes and metabolites in grey were not detected. (**C**) Portal pressures in healthy mice and 7 weeks after induction of ASH or NASH. Data are shown as mean ± SEM (*n* = 10). One-way ANOVA, Tukey post hoc test, * *p* < 0.05, ** *p* < 0.01. NO: nitric oxide; NOS: nitric oxide synthase.

**Figure 7 ijms-24-10351-f007:**
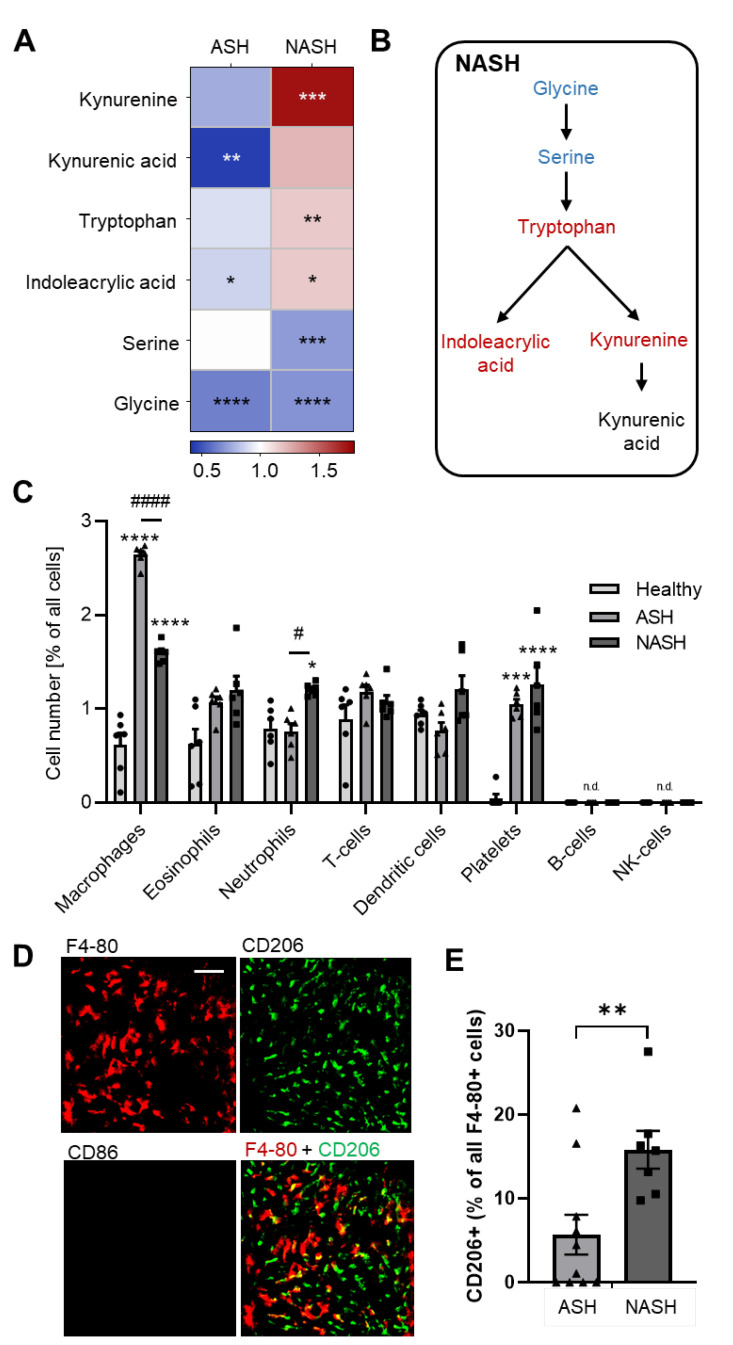
Distinct immune responses in liver samples of mice with induced ASH and NASH after 7 weeks. (**A**) Heatmap of the detectable metabolites of the tryptophan metabolism in liver tissues 7 weeks after induction of ASH or NASH. Data are shown as fold changes compared to healthy mice (*n* = 8). Student’s *t*-test vs. healthy mice, * *p* < 0.05, ** *p* < 0.01, *** *p* < 0.001, **** *p* < 0.0001. (**B**) Schematic illustration showing the regulated metabolites of the tryptophan metabolic pathway in liver samples of mice with NASH. Metabolites shown in blue are significantly downregulated, whereas metabolites in red are significantly upregulated as compared to healthy animals. Metabolites indicated in black show no significant changes. (**C**) Immune cell counts were determined using high-content immunohistochemistry in liver tissues from healthy mice and 7 weeks after induction of ASH or NASH. Data are shown as mean ± SEM (*n* = 6). Two-way ANOVA, Tukey post hoc test, * *p* < 0.05, *** *p* < 0.001, **** *p* < 0.0001 vs. healthy mice, # *p* < 0.05, #### *p* < 0.0001 ASH vs. NASH. n.d., not detected. (**D**,**E**) Representative images (panel **D**) and quantification (panel **E**) of M-like macrophages (F4 80^+^/CD86^−^/CD206^+^) in liver tissue 7 weeks after induction of ASH or NASH. Data are shown as mean ± SEM (*n* = 6). Unpaired Student’s *t*-test, ** *p* < 0.01. The white bar represents 50 mm.

## Data Availability

All data are available in the main text or the Appendix A.

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
