# Peer review of "Differential Lipidomics, Metabolomics and Immunological Analysis of Alcoholic and Non-Alcoholic Steatohepatitis in Mice"

_ijms, 2023, doi:10.3390/ijms241210351_

Round 1
Reviewer 1 Report
This reviewer found this article delightful. It is well written, clear and objective. The goal is also interesting and results quite unexpected in many ways.
I would like to point it out just a few curiosities:
- Why did the authors use differente post-hoc tests afetr ANOVA? Why not stick to one?
-Although the authros have stated this is an established model for ASH and NASH, how does a CCL4 injected only mice behave? What changes are expected? This should be at least adressed in discussion.
Author Response
Why did the authors use different post-hoc tests after ANOVA? Why not stick to one?
The tests were used by the different collaborating groups, but were harmonized in the revised version of the manuscript for using Tukey as post hoc test for all One-way ANOVA analyses. There were no relevant changes in the significance levels, which would affect the overall conclusions. In Fig. 1E the difference between the healthy and the NASH group is no longer significant, while the difference between ASH and NASH or healthy mice remains significant.
We changed the figures and the text throughout the manuscript to include the changes of post hoc tests.
Although the authors have stated this is an established model for ASH and NASH, how does a CCL4 injected only mice behave? What changes are expected? This should be at least addressed in discussion.
As requested we added a paragraph to the “Discussion” section that describes the effect of CCL4 Injection and the advantage of the combined treatments with CCL4 and western diet/ethanol as follows (page 13):
“As animal models co-treatment of mice with ethanol and CCl4 for ASH induction or west-ern diet and CCl4 for NASH induction was chosen [12]. Mice receiving CCl4 alone show in absence of fatty liver disease an induction of liver fibrosis as well as reduced body and liver weight. The combination of CCl4 with western diet enhances the full spectrum of fibrosis, steatosis and inflammation including collagen accumulation, hepatic stellate cell activation and hepatic steatosis [12]. The two animal models for ASH and NASH had overall similar disease severity, although some distinct features between the models were seen.”
[12]: Brol MJ et al., Combination of CCl4 with alcoholic and metabolic injuries mimics human liver fibrosis. Am J Physiol Gastrointest Liver Physiol. 2019 Aug 1;317(2):G182-G194. doi: 10.1152/ajpgi.00361.2018
Reviewer 2 Report
Dorochow and Kraus et al. describes the lipidomic and metabolic profiles of mouse models of NASH and ASH, and explores the differences between these two disease models in comparison to healthy control mice. In NASH, metabolic factors and systemic inflammation play a significant role in disease progression, while in ASH, alcohol metabolism, oxidative stress, and direct toxic effects of alcohol metabolites on the liver contribute to the pathology. The work is well presented and the results explained with a nuanced approach, highlighting that while NASH an ASH may present with similar histopathological parameters, their differing etiologies play a major role in how liver injury begins and inflammation/fibrosis progresses. I recommend that the article be published with minor revisions.
Major comments
1. Animal models often have translatability issues when it comes to comparisons with human pathology. It would be great if the authors can comment on the findings of the animal models’ multi-omic characterization in light of what these findings would mean to human disease.
2. Platelets counts were found to be higher in the disease tissues (Fig 7C). Please include this result in the section 2.4, and comment on the roles of platelets in immune recruitment and immune/HSC inflammatory activation (doi: 10.3389/fphar.2022.842636). Also please correct the typo in Fig 7C – “platelets” is spelt wrong.
3. Please comment on the implications of the absence of CD86+ (M1) macrophages in the disease tissue. M1 macrophages are generally known to be important to NASH and ASH inflammation, and their absence is quite surprising.
4. Have the authors looked at the aspects of mitochondrial dysfunction between the three treatment groups? Genes like SIRT-1 and PGC-1α are often differentially expressed in NASH/ASH diseases, and it would be interesting to see the differences in gene expression of mitochondrial enzymes that are downregulated in the event of mitochondrial dysregulation. RNA/cDNA samples used for panels 1G/1H may be used to answer this question.
Minor comments
1. In Fig 1I, the value depicted in the bargraph is the measured autofluorescence per unit area. It is therefore inaccurate to describe this measure as “hepatic stellate cells/unit area”.
2. Methods section to be updated to include portal pressure measurement methodology.
3. Figure 7D – Please indicate in the figure or the legend if the images are from NASH or ASH samples. Also, line 307 to be corrected to read “M2-like macrophages”.
4. Fig 7D - It would also be useful to include the representative images for F4-80/CD206/CD86 staining of the disease and control samples, perhaps as a supplemental file.
5. Various figures refer to the same treatment group with different names – “naïve”, “control”, and “healthy”. Please change them to be consistent throughout the manuscript.
Author Response
Major comments
- Animal models often have translatability issues when it comes to comparisons with human pathology. It would be great if the authors can comment on the findings of the animal models’ multi-omic characterization in light of what these findings would mean to human disease.
To answer this important comment, we added a paragraph to the “Discussion” section addressing the comparability of the ASH and NASH models with findings in human cohorts.
“The CCl4/ethanol and CCL4/western diet mouse models compared the transcription levels of markers of fibrosis, steatosis, inflammation, and proliferation. Transcript correlation and PC analysis matched well between the CCl4/ethanol model and a human alcoholic liver cirrhosis cohort as well as the CCl4/WD model and a human NAFLD cohort, confirming etiology-specific modeling in the mice models [12]. More recent publications were able to correlate effects of Janus kinase 2 inhibitors [24] as well as the role of macrophage-inducible C-type lectin (mincle) [25] in these models with results from human cohorts.”
- Torres, S.; Ortiz, C.; Bachtler, N.; Gu, W.; Grünewald, L.D.; Kraus, N.; Schierwagen, R.; Hieber, C.; Meier, C.; Tyc, O.; et al. Janus kinase 2 inhibition by pacritinib as potential therapeutic target for liver fibrosis. Hepatology 2023, 77, 1228–1240, doi:10.1002/hep.32746.
- Schierwagen, R.; Uschner, F.E.; Ortiz, C.; Torres, S.; Brol, M.J.; Tyc, O.; Gu, W.; Grimm, C.; Zeuzem, S.; Plamper, A.; et al. The Role of Macrophage-Inducible C-Type Lectin in Different Stages of Chronic Liver Disease. Front. Immunol. 2020, 11, 1352, doi:10.3389/fimmu.2020.01352.
- Platelets counts were found to be higher in the disease tissues (Fig 7C). Please include this result in the section 2.4, and comment on the roles of platelets in immune recruitment and immune/HSC inflammatory activation (doi: 10.3389/fphar.2022.842636). Also please correct the typo in Fig 7C – “platelets” is spelt wrong.
As requested we mention the platelet data in the “Results” (page 11) and added a paragraph to the “Discussion” (page 14) section of the manuscript as follows:
Notably, leukocytes can transport platelets in inflamed tissues where the platelets pro-mote inflammatory processes [33–35]. Since we found similar platelet numbers in liver tissue from mice ASH and NASH, indicating that neutrophils replace macrophages in the hepatic platelet transport. Once in the tissue platelet can be activated and suppress expression of markers for M2-like macrophages [34] and to activate HSCs through a variety of mediators [33].
[33]: Dalbeni et al., Platelets in Non-alcoholic Fatty Liver Disease. Front. Pharmacol. 2022, 13, 842636, doi:10.3389/fphar.2022.842636.
[34]: Pierre et al., GPVI and Thromboxane Receptor on Platelets Promote Proinflammatory Macrophage Phenotypes during Cutaneous Inflammation. J. Invest. Dermatol. 2017, 137, 686–695, doi:10.1016/j.jid.2016.09.036.
[35]: Malehmir et al., Platelet GPIbα is a mediator and potential interventional target for NASH and subsequent liver cancer. Nat. Med. 2019, 25, 641–655, doi:10.1038/s41591-019-0379-5.
- Please comment on the implications of the absence of CD86+ (M1) macrophages in the disease tissue. M1 macrophages are generally known to be important to NASH and ASH inflammation, and their absence is quite surprising.
We agree with the reviewer that the absence of CD86-positive macrophages is surprising, since these are the prototypical pro-inflammatory macrophages found in many inflammation models and could be expected in inflamed livers. However, while we can rule out technical problems with CD86 detection due to positive staining of control tissues, we can not rule out that there are specific effects on hepatic macrophage polarization in the liver disease models, for example in regard to the CCl4 treatments.
- Have the authors looked at the aspects of mitochondrial dysfunction between the three treatment groups? Genes like SIRT-1 and PGC-1α are often differentially expressed in NASH/ASH diseases, and it would be interesting to see the differences in gene expression of mitochondrial enzymes that are downregulated in the event of mitochondrial dysregulation. RNA/cDNA samples used for panels 1G/1H may be used to answer this question.
As suggested by the reviewer we determined the mRNA expression levels for SIRT-1 and PGC-1α. We found no significant change for SIRT-1 or PGC-1α mRNA expression in ASH as compared to healthy mice. However, SIRT-1 mRNA expression was significantly reduced in NASH as compared to healthy mice or mice with ASH. These findings support our other data suggesting a mildly increased disease progression in NASH.
The data have been included in the revised manuscript as Figure S1
Minor comments
- In Fig 1I, the value depicted in the bargraph is the measured autofluorescence per unit area. It is therefore inaccurate to describe this measure as “hepatic stellate cells/unit area”.
The labelling of the y-axis in Figure 1I was changed as requested.
- Methods section to be updated to include portal pressure measurement methodology.
We apologize for the missing method description and added it to the revised manuscript.
- Figure 7D – Please indicate in the figure or the legend if the images are from NASH or ASH samples. Also, line 307 to be corrected to read “M2-like macrophages”.
The images were from mice with ASH. We changed the figure legend accordingly.
- Fig 7D - It would also be useful to include the representative images for F4-80/CD206/CD86 staining of the disease and control samples, perhaps as a supplemental file.
As requested by the reviewer we added the images to the manuscript as Figure S5.
- Various figures refer to the same treatment group with different names – “naïve”, “control”, and “healthy”. Please change them to be consistent throughout the manuscript.
The control groups description was changed throughout the manuscript to “healthy”.